# Mitigation Strategies against Food Safety Contaminant Transmission from Black Soldier Fly Larva Bioconversion

**DOI:** 10.3390/ani14111590

**Published:** 2024-05-28

**Authors:** Matan Shelomi

**Affiliations:** Department of Entomology, National Taiwan University, 106319 Taipei, Taiwan; mshelomi@ntu.edu.tw; Tel.: +886-0233665588

**Keywords:** circularity, food safety, heavy metals, *Hermetia illucens*, microbial hazards, pathogens, toxins

## Abstract

**Simple Summary:**

Larvae of the black soldier fly can eat organic waste and grow into nutritious sources of protein and fat for animal feed. However, if the waste they eat is contaminated with toxins or dangerous microbes, a risk exists that the larvae will become contaminated as well. This review examines the research on ways to decontaminate waste before feeding it to larvae or decontaminate larvae after feeding them waste to identify the safest way to minimize the risk to consumers from such hazards. Sample treatments include using heat or disinfectants; however, these can affect larval growth rates or the quality of the final product. Using less contaminated wastes may be required for black soldier fly larvae destined for use as animal feed, whereas larvae fed highly contaminated wastes can be directed to non-feed uses.

**Abstract:**

The black soldier fly larva, *Hermetia illucens*, can efficiently convert organic waste into biomatter for use in animal feed. This circularity comes with a risk of contaminating downstream consumers of the larval products with microbes, heavy metals, and other hazards potentially present in the initial substrate. This review examines research on mitigation techniques to manage these contaminants, from pretreatment of the substrate to post-treatment of the larvae. While much research has been done on such techniques, little of it focused on their effects on food safety contaminants. Cheap and low-technology heat treatment can reduce substrate and larval microbial load. Emptying the larval gut through starvation is understudied but promising. Black soldier fly larvae accumulate certain heavy metals like cadmium, and their ability to process certain hazards is unknown, which is why some government authorities are erring on the side of caution regarding how larval bioconversion can be used within feed production. Different substrates have different risks and some mitigation strategies may affect larval rearing performance and the final products negatively, so different producers will need to choose the right strategy for their system to balance cost-effectiveness with sustainability and safety.

## 1. Introduction

The last decade has seen an ongoing global surge in the establishment of companies or research teams investigating the use of black soldier fly (*Hermetia illucens*) larvae, or BSFL, for use in the circular economy [1]. BSFL are capable of converting a wide diversity of organic waste into biomass at a high rate, more so than other fly larvae like house flies [2]. The resulting larval, prepupal, or pupal meal can replace soymeal or fishmeal in the feed for livestock such as chicken or fish up to 100% with no negative effects on the animal or the final human consumer [3]. Thus, BSFL are seen as a critical component to closing nutrient loops in food production: organic side streams and pre- or post-consumer wastes can be fed to the flies, and the flies fed to livestock whose subsequent wastes, including both manure and slaughterhouse waste, could potentially also be used to feed the BSFL [4,5].

While this circularity has high potential for increasing the sustainability of both the food used as a substrate and the food fed the BSFL, it also poses a food safety issue in the transfer of pathogens and toxins from the former to the latter through the flies [6]. Some preliminary work suggested that flies reduce the contaminant load of the substrates they feed on by digesting and destroying them, to the point that BSFL treatment of certain types of waste is directly beneficial as a means of decontamination before the waste is used for other purposes such as composting [7]. Other researchers have looked into what compounds or elements BSFL do and do not bioaccumulate and what pathogens can travel in their gut. The hope from the standpoint of sustainability is that BSFL can be fed otherwise toxic wastes and simultaneously decontaminate them and convert them into safe animal feed [8]. The reality will likely be that certain substrates pose an unacceptable risk to consumers if used to feed BSFL that enter the food chain, and conservative legal bodies may formalize the limitations of what can be fed to BSFL and what BSFL can subsequently be fed to [3].

This narrative review [9,10] seeks to consolidate extant information on mitigating the risks of contamination transfer from BSFL. These strategies can be divided into pretreatment of substrates before BSFL bioconversion, post-treatment of the larvae or selection of certain life stages before subsequent usage as animal feed, and legal limitations on the types of substrates used for BSFL feeding and types of downstream applications for the BSFL themselves (Figure 1). For this type of review, the PRISMA guidelines and preregistration of protocols are neither required nor desired or appropriate [9,11,12]. Nonetheless, a protocol was registered at https://osf.io/jfwv4/ (accessed on 26 May 2024). Database searches of Google Scholar were used for articles with “black soldier fly” and other relevant key terms relating to heavy metals, toxins, pathogens, pesticides, pharmaceuticals, and other contaminants, as well as certain treatment types, in the title, abstract, or keywords, followed by snowballing with references from these papers’ citations (which previous work confirms is more than sufficient for such reviews) [13]. Narrative reviews are by definition noncomprehensive and non-exhaustive [14], so this review is limited to research and review articles on BSFL production published or preprinted in peer-reviewed journals or as academic theses.

## 2. Background

### 2.1. Microbial Hazards

Until 2023, no major disease outbreaks had been reported in any BSFL rearing center, while several have occurred in farms for other mass-produced insects [15]. To date, only one case of infectious disease among farmed BSF has been reported: “soft rot” involving the *Paenibacillus thiaminolyticus* [16]. Particularly strong immunity is inherent in the species [17]: detritivorous and saprophagous insects like BSFL need to have evolved systems to decontaminate the decaying substrates on which they feed, such as by producing antimicrobial peptides [18]. The BSFL gut varies in pH along its length, from 5.9 ± 0.1 in the anterior midgut to 2.1 ± 0.1 in the middle midgut and 8.3 ± 0.2 in the posterior midgut [19], which is similar to what has been observed in house fly and fruit fly larvae [20]. Extreme pH, along with protective enzymes, antimicrobial peptides [21], and microbial antagonism from their gut microbes [22,23], may kill bacteria like *Escherichia coli*, *Staphylococcus aureus*, and *Salmonella* spp., even from substrates as contaminated as fecal sludge and manure [24]. Not only would the resultant larvae be uncontaminated, but also the substrate would be decontaminated or “hygienized” by the larval activity [20,23,25,26,27,28,29,30,31]. Reductions in substrate pathogen loads may be caused not only by direct destruction in the BSFL gut or secreted antimicrobial compounds [31] but also by BSFL gut microbes colonizing the substrate and outcompeting the pathogens [23], with evidence suggesting the latter is the main mechanism for BSFL’s pathogen-reducing ability [32]. However, BSFL is not perfect, with one study finding no effect of BSFL on substrate *Salmonella* levels [33] and another finding no effect on certain livestock parasites [34]. *Salmonella* and *Listeria monocytogenes* are not typically found in BSFL [6,35,36], but a significantly contaminated substrate or a particularly resistant or competitive strain could overwhelm the fly’s antimicrobial abilities [33], and pathogens like *Salmonella* and *Bacillus cereus* have been found in BSFL bioconversion residue even when not detected in the larvae themselves [37]. Most authors err on the side of caution and say pre-bioconversion decontamination of the substrate and post-harvesting decontamination of BSFL and of the residue left after bioconversion should always be used [28,33,37].

Evidence for nonbacterial pathogens is more limited. Studies on porcine respiratory coronavirus and African swine fever virus found that the virus is detectable in the larvae for up to three days post-exposure, but pigs fed unprocessed larvae were not subsequently infected [38,39]. BSFL have been found to completely eliminate all mycelial fungi from food waste [40], and artificially synthesized BSFL antimicrobial peptides are effective against phytopathogenic fungi [41]. BSFL do not seem to reduce helminth numbers in waste [42,43]. Little to no data exist for protozoa transfer via BSF, and none at all for prions [3]. Note that the possibility always exists that a novel BSF disease can emerge in the future [44], as for any other organism.

### 2.2. Nonmicrobial Hazards

BSFL do bioaccumulate heavy metals and other elements depending on what is present in the feed, meaning some studies find no bioaccumulation of certain elements that other studies do. The bioaccumulation factors for cadmium and manganese are consistently the highest, followed by arsenic, lead, iron, zinc, mercury, copper, chromium, magnesium, potassium, and tin; these factors are consistently below 1 (no bioaccumulation) for aluminum, cobalt, molybdenum, nickel, selenium, silver, or vanadium [6,45,46,47,48,49,50,51,52,53,54,55,56,57,58]. While this could reduce the suitability of the insects for use in feed, it does serve to decontaminate the substrate and produce a cleaner residue for downstream uses [8,54,57]. Some evidence suggests heavy metal contamination negatively affects the natural BSFL gut microbiome and leads to the enrichment of pathogens in the gut [56,59].

Among other chemical contaminants, BSFL were found to accumulate mineral oil and some other hydrocarbons but none exceeding legal limits [47]. In general, BSFL are not considered to accumulate dioxins, polychlorinated biphenyls, or polycyclic aromatic hydrocarbons, and they do not accumulate and, at times, they even degrade many pharmaceuticals and pesticides [7,49,50,60,61,62,63,64]. One study found that substrate contaminated with malathion did affect BSFL negatively, but that study did not assess the impact of BSFL on the malathion levels in the residue or the prepupae [65]. Another found that λ-cyhalothrin negatively affected BSFL development but was nonetheless degraded in the substrate and not bioaccumulated [66]. Others found that while spinosad, deltamethrin, and cypermethrin present in a substrate at the EU’s maximum residue level negatively affected BSFL growth, imidacloprid stimulated growth while chlorpyrifos, propoxur, and tebufenozide had no effect even at ten times the maximum residue level [64,67]. Toxicity increased if multiple compounds were in the substrate at the same time, or if synergists like piperonyl butoxide were added. Prior work suggests BSFL have high tolerance to aflatoxins and other mycotoxins and do not bioaccumulate them [49,51,60,68,69,70] though some may persist in their gut contents. While some plant toxins, like genotoxic pyrrolizidine alkaloids (PAs) and tropane alkaloids, were degraded from the substrate during BSFL bioconversion, others bioaccumulated in the larvae at varying rates: high for europine, rinderine and echinatine and very low for most other PAs and atropine and scopolamine [71].

Like many insects, BSFL have crustacean cross-reactive allergenic proteins like tropomyosin, arginine kinase, and myosin that are inherent risks in the insect tissue and likely cannot be removed without post-harvesting chemical processing [6,72].

## 3. Pretreatment of Substrate before Bioconversion

Pretreatment options for substrates include mechanical (milling, disintegration), thermal, radiological, chemical (addition of oxidizing agents, acids, alkalis, ozone, biochar, and enzymes), and biological (fermentation with lignocellulase-producing microbes, co-treatment of microbes together with BSF) treatments [73,74,75,76,77]. Studies of these substrate treatments are typically conducted with the goal of optimizing BSFL bioconversion and growth rates [78], such as by digesting recalcitrant lignocellulosic material [74] rather than improving food safety. Limited studies thus exist on how these pretreatments affect pathogen or chemical contamination of the substrate or subsequent larval biomass.

The few studies that have been conducted confirm that heat pretreatment can effectively reduce microbe counts in substrates such as fecal sludge and supermarket waste without subsequent negative effects on the larvae relative to untreated substrate [79,80]. Different wastes will require different time–temperature combinations: for example, 10 min at 60 °C was enough to eliminate deliberately inoculated *Salmonella* and *Staphylococcus aureus* from meatless supermarket waste, which could then be stored at ambient temperature without microbe growth for over a day [80]. Some studies warned that thermal pretreatment can reduce the presence of helpful microbes in a substrate as well as increase the presence of tannins and phenolic compounds that are toxic to the larvae [73,81] though other studies found beneficial effects of pretreatment [82], so ultimately the effects will vary with the substrate. Dry substrate typically has lower counts of pathogenic microbes like *Staphylococcus aureus* [28], and research unrelated to BSFL also suggests drying can affect heavy metal speciation and stability [83], but the impact of drying a substrate is limited as BSFL require a moist substrate to survive [84].

Ammonia pretreatment of the substrate was sometimes found to be toxic to BSFL [85] or their gut microbes [86] but not consistently [87], and no papers examined its effects on contamination. The use of disinfectants against African swine fever virus in pig manure had varying effects: BSFL growth and bioconversion rates were higher than the control when exposed to potassium peroxymonosulfate and lower than the control when exposed to glutaraldehyde, plus both had unexpectedly positive effects on the BSFL microbiome diversity [88]. Irradiating food waste prior to feeding it to BSFL deactivated the original microbiota but also decreased all measures of larval rearing performance [77]. Grinding food waste substrate into particles smaller than 2 mm was found to negatively affect bioconversion and correlated with an increased prevalence of potentially entomopathogenic *Morganella* in the BSFL gut though that strain was not thought to have been acquired from the substrate [89]. Rather, *Morganella* has been repeatedly found in BSFL guts and is considered to be part of the typical BSFL gut microbiome, along with *Dysgonomonas, Enterobacter*, *Enterococcus*, *Klebsiella*, *Lactobacillus, Proteus*, and *Providencia* [77,90,91,92,93,94]. *Morganella* is known to increase in prevalence in the BSFL gut when the diet is contaminated with oxytetracycline exposure [63]. Other gut microbes, including *Dysgonomonas* and *Enterococcus,* were found to be responsible for the sulfadiazine in the gut [95]. Manipulating the pH of the initial substrate before adding BSFL was not found to affect BSFL’s suppression of pathogenic microbes from the substrate, nor did it affect the gut microbiome of the BSFL [23]. That recent report suggested that BSFL gut microbes like *Lactobacillus* spp. grow in the substrate and outcompete pathogens while pathogenic microbes do not colonize the BSFL gut. However, further studies are needed to test a wider array of contaminated substrates [23]. As for abiotic hazards, the co-application of *Lactobacillus buchneri* with BSFL to the substrate had no effect on cadmium or chromium levels [96].

## 4. Post-Treatment of Larvae after Bioconversion

For edible insects, as with other foods, several post-harvesting strategies exist that can minimize contamination with pathogens; although, again, the majority of research on these techniques has focused on factors like digestibility, nutrition, and stability rather than safety. A project on edible insects in Kenya found that raw insects were often contaminated with *Salmonella* to unacceptable levels, requiring some form of cooking to make them safe for human consumption regardless of the effects on nutrition [97]. For BSF, the process starts with the method of killing, with the exceptions of cases where the live larvae are fed to fish or poultry in the vicinity of the bioconversion facility. Blanching, boiling, toasting, desiccation, freezing, high hydrostatic pressure processing, asphyxiation, and grinding have been tested for their effects on subsequent microbial load, though one must note all can affect the final consistency, digestibility, and nutritional and organoleptic properties of the BSFL product, which are factors producers will have to take into account [35,98,99,100].

Comparative studies of these factors found blanching to have superior antimicrobial effects, as well as minimal negative effects on other aspects of the product like lipid oxidation or digestibility and a positive effect on initiating dehydration in BSFL [6,98,101,102,103]. Different authors who tested blanching BSFL for the same amount of time (40 s) nonetheless found different levels of microbes, from undetectable [98] to high and requiring further processing [102,103]. Blanching can effectively reduce total viable counts of *Escherichia coli, Salmonella,* and vegetative cells of *Bacillus cereus* but does not significantly reduce *Bacillus cereus* spores [104] or other endospore formers like *Clostridia*. Some studies suggest that a killing method also has variable effects on the heavy metal content of the larvae [6]. A disadvantage of killing by blanching is that it may be painful to the fly larvae: heat is known to induce nociceptive behavior in Diptera [105]. Grinding is generally accepted as the most humane way to slaughter large numbers of insects [106], but as it does not minimize microbial contamination [98], it would require prior or subsequent decontamination strategies such as heat treatment or lactic acid fermentation [107].

As much of the microbial contamination of BSFL is expected to be in the gut contents, and as these are nutrient poor relative to the fat- and protein-rich insect tissue, gutting [actively removing the digestive tract] or “gastrointestinal evacuation” of the larvae through pre-killing starvation should reduce contamination and improve the quality of the final product [108,109]; however, there may be a large difference in effectiveness between gutting and starvation. A study found maintaining BSFL larvae for 6 h in water or 48 h without food led to clearing of the gut with improved organoleptic and nutritional qualities and reduced microbial load, so this could be a suitable addition to a harvesting program for BSFL at the cost of increased time between harvest and processing and possible ethical concerns for BSFL welfare [110]. Note too that a similar study with *Tenebrio molitor* found no effect of starvation on microbial numbers or bacterial community composition [111].

The other part of the BSFL expected to be contaminated with microbes is, unsurprisingly, the surface. Washing the larvae to remove the leftover residue, which may be commonly performed as a routine part of harvesting, could also remove certain contaminants, but the degree to which they are removed has barely been studied. Washing was found to be ineffective at removing the persistent parasite stages (oocysts and eggs) of coccidian parasites *Eimeria nieschulzi* and *Eimeria tenella* or the nematode parasite *Ascaris suum* that attaches to the larval surface [34], posing a risk to humans and animals such as chickens and pigs if the harvested BSFL is not further treated with heat or ammonia. Note too that these parasites were not destroyed by the BSFL gut and passed through it undigested [34].

Post killing, BSFL like any other edible insect can be desiccated with sunlight, hot air, infrared or microwave radiation, freeze-drying, or use of pulsed electric fields; it can also be frozen, pulverized, milled, pressed, fractionated, blanched in water or acid solutions, pasteurized, toasted, smoked, fried, and/or fermented [99,102,112,113,114,115]. In addition to reducing microbial contamination, these techniques reduce water content and can improve shelf life and stability [116]. As before, however, the majority of the research on post-killing treatments focus on the energy costs of the drying method or the nutritional value, digestibility, and organoleptic properties of the BSFL product rather than the safety [115,117,118]. Combinations of these steps can be performed in a different order, with varying effects on the final product from both a consumer and a safety point of view [102]. For example, boiling thawed BSFL for four minutes followed by six hours of hot air drying at 60 °C and then grinding into powder produced a stable product with no detectable *Salmonella* or *E. coli* [102]. The costs also vary immensely, and different methods will be employed on a large scale, such as industrial BSFL bioconversion facilities in a wealthy nation versus a household scale setup in the developing world [119,120]. A comparison of oven drying and sun drying found that while oven drying is superior, both were acceptable in reducing pathogenic bacteria counts so long as the substrate used for the BSFL production was not overly contaminated [121]. In addition, fractionation of BSFL into separate protein, oil, and chitin components could create products with different risk profiles. For example, bioaccumulated metals were not always present in pressed larval oil from BSFL fed sludge with heavy metals [122].

Much of the data on post-treatment of insects on microbial contamination looked at species grown for human consumption, such as *Tenebrio molitor*, rather than BSF. While some of the data can be extrapolated to BSF, these papers technically fall outside the purview of this review. A few papers examined post-treatment of killed BSF. In one, puncturing larvae to speed their drying increased microbial load compared with unpunctured larvae, while blanching or scalding in boiling water reduced microbial load while still speeding up the drying process [102]. In another, both infrared-assisted air drying and drying with a pulsed electric field reduced fungi, Enterobacteriaceae, spore-forming bacteria, and lactic acid bacteria counts significantly but not sufficiently to bring them below unsafe levels for food purposes [112]. A study on four African edible insects including BSFL found that boiling and toasting completely or significantly reduced bacterial and yeast populations, oven drying marginally reduced them, and solar drying had no effect [123]. A paper on lactic acid fermentation of BSF prepupa, puparia [abandoned pupal exuviae], and dead adults using *Lacticaseibacillus rhamnosus* 1473 and *Lactiplantibacillus plantarum* 285 found puparia fermented with either microbe and dead adults fermented with the latter had significant antimicrobial effects on *Listeria monocytogenes*, *Salmonella*, and *Escherichia coli* [124]. Waste-fed BSF prepupa killed by refrigeration for one day and sun-dried for one week before being powdered and pelletized into fish feed had microbial and heavy metal levels well below EU regulations [125]. The literature suggests expensive, novel technologies like pulsed electric field drying may not be superior to traditional methods like cooking.

After harvesting and processing, BSFL products could then be packed, stored, and distributed to final consumers. Vacuum packaging of living, blanched, and frozen larvae produced microbial loads identical to chilled larvae stored in air [126]. Studies comparing BSFL powder stored in woven polypropylene sacks, polyethylene bags, and screw-lidded plastic containers found that microbial degradation, including *Salmonella* growth, was fastest in the bags and slowest in the plastic container, with refrigeration usually slowing microbe growth but sometimes exacerbating spoilage in the woven bag due to moisture [127,128]. One should remember that general food safety issues apply to edible insects as they do to any other food: at any point during shipping and handling and final consumption, the BSFL product can become contaminated with microbes even if none were present at the point of harvest or after initial processing [129].

## 5. Legal Restrictions on BSFL Bioconversion

Different governments have different laws regarding what kinds of feed can be fed to what kind of insect, what kinds of insects can be fed to what kinds of livestock, whether and how insects can be used as human food, and what kind of processing is allowed or required for insect matter to be used as food or feed [130,131], although the majority of nations have no such laws, guidelines, or regulations [132]. The USA allows for dried BSFL-fed “feed grade materials” to be used in food for salmonid fishes only [3]. The European Union’s Regulation (EC) No 767/2009 and Regulation (EC) No 1069/2009 prohibit the use of manure or any digestive tract content or excrement as insect feed, as well as any food or catering waste that may contain meat or fish, pending further research on the fates of contaminants and allergens on the chain from substrate to insect to livestock to human [109,133]. Similarly, processed animal protein from certain insects (including BSFL) can only be used for fish, pigs, and poultry but not ruminants like sheep or cattle [133]: a restriction that arose from the catastrophic effects of the 1990s bovine spongiform encephalitis outbreak attributed to the use of processed animal protein and the inability to detect the animal of origin of processed animal protein to preclude intra-species recycling [134]. These restrictions do not apply to extracted insect fats or hydrolyzed insect proteins, both of which follow similar regulations as other animal fats and hydrolyzed animal proteins. Live BSFL can be fed to fish, chickens, and pigs with few restrictions as neither are known to get prion diseases [135], but BSFL cannot be fed to susceptible cows or sheep [136,137], and whole dead BSFL can only be fed to pets or zoo animals. While insects do not have prion diseases and prions cannot replicate in insect cells [138], prions are “sticky” proteins, which are difficult to eliminate, and could physically contaminate the gut or surface of a BSF prepupa [139,140]. As prion diseases like bovine spongiform encephalitis are untreatable, incurable, and invariably fatal, the lack of data on the fate of prions during BSFL bioconversion means restrictions relating to food waste as substrate and fly protein in non-fish feed are unlikely to be lifted until the question of whether the fly can vector prions is solved and perhaps not even then.

## 6. Conclusions

The review into strategies to mitigate food safety risks from BSFL bioconversion found a relative paucity of the literature on the effects of substrate pretreatment and insect post-treatment on microbial and chemical contamination despite an abundance of research on pretreatment and post-treatment for other variables, like bioconversion efficiency, digestibility, and nutrition, respectively. In the absence of data on risks and safety, governments like the EU will take a conservative approach and restrict the types of waste that can be given to BSFL destined for use as feed, which, in turn, greatly limits the suitability of BSFL to reduce waste and produce feed sustainably [141,142]. Future works on pretreatment should thus include a microbial component comparing contamination before and after treatment to fill this gap in the literature. Results regarding post-harvesting treatment in other foods may well apply to black soldier fly, but for food regulatory bodies in the EU, data specifically produced on *Hermetia illucens* is ideal.

Control or decontamination of substrates before BSFL application, for example with heat treatment, can be an effective way to minimize contamination of subsequent BSFL products. Killing larvae by blanching before subsequent freeze-drying or dehydrating is a commonly reported method for minimizing microbial contamination, but the optimal protocol likely needs to be determined on a case-by-case basis. Gastrointestinal evacuation before killing appears promising and needs significantly more research. BSFL do bioaccumulate certain heavy metals, which can render them unusable as feed but do serve to decontaminate the substrate and leave a usable residue. For producers, a balance will need to be struck between decontaminating the substrate without negatively affecting BSFL performance and between decontaminating the larvae without negatively affecting the final product. The ideal balance will depend largely on the type of substrate and its inherent risks, as well as the proposed final form of the BSFL product. In developed nations, legal and ethical issues may need to be considered, while in developing economies issues of cost and available technology will matter more. The highest contaminants of concern are metals like cadmium, endospore-forming microbes, parasites, and prions. Certain hazards such as prion contamination must be controlled for at the substrate level by decontamination or, until available data suggests alternatives, restrictions on substrates. Black soldier fly larvae employed primarily to eliminate hazardous wastes for which decontamination methods are insufficient or impractical could also be directed to non-feed uses such as chemical or biodiesel production [20,143].

Finally, note that BSFL production itself produces undesirable wastes, including greenhouse gases, ammonia, and nitrous oxide [144], though arguably at lower rates than certain other waste management techniques or biomatter production methods. BSFL bioconversion produces odors that may necessitate management such as using air scrubber to minimize the smells coming from a fly facility, though the odors from BSFL production are generally lower than allowing wastes to decay [145]. Finally, BSFL production leaves behind undigested substrate, frass, exuviae, and dead adults, though the management and even valorization of these wastes is the subject of other reviews [146,147,148].

## Figures and Tables

**Figure 1 animals-14-01590-f001:**
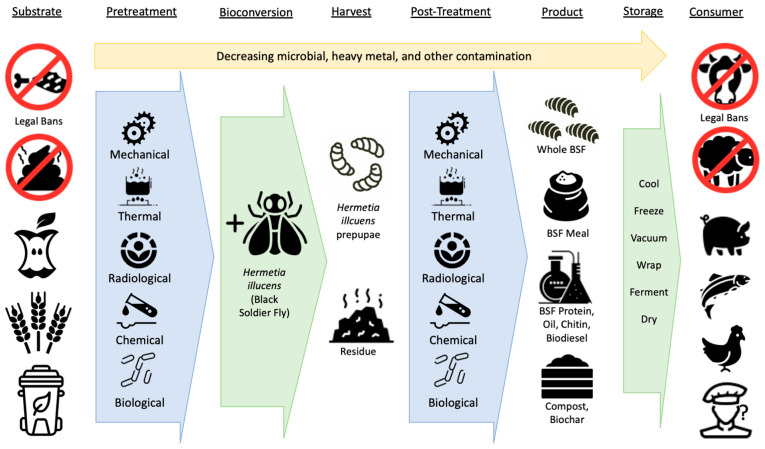
Pre- and post-bioconversion mitigation strategies to reduce contamination in black soldier fly products.

## Data Availability

No new data were created or analyzed in this study. Data sharing is not applicable to this article.

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
