# Peer review of "Mitigation Strategies against Food Safety Contaminant Transmission from Black Soldier Fly Larva Bioconversion"

_animals, 2024, doi:10.3390/ani14111590_

Round 1

Reviewer 1 Report

Comments and Suggestions for Authors

This review provides an overview of the process of utilizing black soldier fly larvae to convert organic waste into animal feed, as well as the associated risks. It reviewed key issues such as microbial contamination, heavy metals, and other potential pollutants, as well as the research techniques for managing these pollutants. The authors summarize various techniques for mitigating pollution risks, while also highlighting some of the challenges and unknowns involved. In proposing future research directions, the authors suggest a greater focus on analyzing food safety contaminants and studying the effects of different mitigation strategies on production systems. it offers valuable insights for researchers and practitioners in the field, but there are areas where further refinement and deepening are needed.

Lines 84: please list some specific antimicrobial substances from BSFL, such as antimicrobial peptides (AMPs), chitosan and etc as the following references:

1.     Xia J, Ge C, Yao H. Antimicrobial peptides from black soldier fly (Hermetia illucens) as potential antimicrobial factors representing an alternative to antibiotics in livestock farming[J]. Animals, 2021, 11(7): 1937.

2.     Guarnieri A, Triunfo M, Scieuzo C, et al. Antimicrobial properties of chitosan from different developmental stages of the bioconverter insect Hermetia illucens[J]. Scientific Reports, 2022, 12(1): 8084.

Line 182: Recent studies clarified the mechanism of SDZ elimination mediated by the gut microbiota of BSFL using DNA-SIP technology and validated the potential of BSFL in removing antibiotics from contaminated substrates. Please refer to the following literature for more convincing:

1.     Xia J, Ge C, Yao H. Identification of functional microflora underlying the biodegradation of sulfadiazine-contaminated substrates by Hermetia illucens[J]. Journal of Hazardous Materials, 2024, 463: 132892.

This review examines research on mitigation techniques to manage these contaminants, from pre-treatment of the substrate to post-treatment of the larvae. Whether intermediate pollutants are also produced during the process of BSFL bioconversion, and what is the situation? Please add to that.

The topic of the article is “Mitigation strategies against contaminant transmission from black soldier fly larva bioconversion”. the author focuses on analysis of food safety contaminants, But what about the subsequent environmental impact? Please add clarification in this paper. Meanwhile, the title should be more specific. 

Comments on the Quality of English Language

Overall, your English language skills are good.

Author Response

Thank you for the revision! Here is a point by point reply:

Line 84 - I added the mention of antimicrobial peptides, but not chitosan. The text is referring to compounds secreted by BSF that decontaminate their environment, but the chitosan paper talks about the antibacterial properties of chitosan extracted from larvae, which does not fit the context. I was able to include that references elsewhere, however.

Line 182 - Thank you! I added this paper.

-You asked about intermediate pollutants. BSF degradation produces odors, and the one paper I found on the subject stated that BSF reduces the odors produced compared to just letting the waste decay. Anecdotally I can tell you BSF production smells and requires odor management, but I do not think BSF produce any different contaminants than what is present otherwise from decaying waste. The only other waste issue I can think of are the frass, adult corpses, etc. I added discussion of this to the end of the manuscript.

-I am unsure what you mean by "environmental impact," but if you mean greenhouse gases and such, that is outside the scope of this review, and has been covered in other, dedicated reviews. I added some text to the end of the manuscript on the subject and cited a published review on the subject.

-As requested, I changed the title to specify "food safety contaminant"

Reviewer 2 Report

Comments and Suggestions for Authors

This is a well written literature review of mitigation strategies against biological and abiotic contaminants that may restrict the use of BSF larvae in downstream applications such as animal feed.

For specialists in the field it is also good to understand, but I feel that this review may be of high interest also for non-academic readers to serve as a guideline for good manufacturing practice in BSF breeding facilities worldwide. I think for example that the pacing of the text and switches between biotic and abiotic contaminants are at some points to abrupt. I also recommend adding sentences explaining specific techniques. For example, Line 167: "Ammonia pre-treatment(...)" is not really self explanatory, so I would wish to have a more specific wording such as "Ammonia pre-treatment of the substrate(...)".

I recommend to structurize the text more (see below).  I think there is a chapter/title missing. After "1. Introduction", "3. PreTreatment..." follows. There seems to be a chapter starting at Line 78 without header. Probably "2. Background". I strongly recommend including subchapters to concentrate the informations and to allow finding specific informations faster. E.g. for assumed chapter 2: Line 78 "BSF larva immunity and known and potential pathogens of BSF larvae", followed by Line 90 "BSF larvae shape the microbial landscape of their environment" or something similar.

The fungal pathogens (both of animal and human health concern but also possibly thretening BSF facilities) are missing nearly completely, and should be included at least in the paragraph strating at Line 105 - non-bacterial pathogens.

Minor comments:

Line 134: if, not of

Line 181: in the not int he

Line 312: No References cited for ostrich prions and pig prions, please add citation.

Author Response

Thank you for the review!

-I added explaining text like that requested for line 167.

-The heading "2. Background" wasn't missing, it was just easy to overlook as it was right after figure 1 as the last line (77) of page 1. I made Figure 1 slightly bigger to solve that problem and push the heading to the start of the next page.

-The reviewer requested subheadings to help concentrate information. I added them for the background section only, separating microbial and non-microbial hazards. I personally prefer short subheadings and strongly dislike papers that insist on a subheading for each paragraph, so stylistically I opted not to add any more. They were not necessary for section 3 given how short it was, and section 4 is divided into different post-treatment types in each paragraph and does not benefit from subheadings.

-As requested, I softened the transitions between biotic and abiotic in places where it was too abrupt.

-Very little fungal data exists, but I did find two papers I missed before and added them.

Line 134 fixed

Line 181 fixed

Line 312 A citation is added, but the parenthetical was also cut

Round 2

Reviewer 1 Report

Comments and Suggestions for Authors

the author has revised the manuscript according to the first comments and suggestions. The paper meets the requirements for publication.

Reviewer 2 Report

Comments and Suggestions for Authors

The review titled "Mitigation strategies against food-safety contaminant transmission from black soldier fly larva bioconversion" collects and concentrates the available informations on methods to reduce the risks of transmission or enrichment of abiotic and biotic contaminants to BSF products or consumers. As stated in my first reply, the review is of high value for both the scientific and economic BSF community.  I recommend the review to be published.

Minor comment: One typo in line 188, please  delete "degradation" after "sulfadiazine"